# Leprosy perceptions and knowledge in endemic districts in India and Indonesia: Differences and commonalities

**Anna Tiny van't Noordende**[1,2]*, **Suchitra Lisam**[3], **Panca Ruthindartri**[4], **Atif Sadiq**[3], **Vivek Singh**[3], **Miftahol Arifin**[4], **Willem Herman van Brakel**[1], **Ida J. Korfage**[2]

**1** NLR, Amsterdam, The Netherlands, **2** Erasmus MC, University Medical Center, Rotterdam, The Netherlands, **3** NLR India, New Delhi, India, **4** NLR Indonesia, Jakarta, Indonesia

* a.vt.noordende@nlrinternational.org

**Data Availability Statement:** All interviews transcripts and an Epi Info database are available from the infolep website: https://www.leprosy-

## Abstract

### Background

Understanding how knowledge, attitudes and practices regarding leprosy differ in endemic countries can help us develop targeted educational and behavioural change interventions. This study aimed to examine the differences and commonalities in and determinants of knowledge, attitudes, practices and fears regarding leprosy in endemic districts in India and Indonesia.

### Principle findings

A cross-sectional mixed-methods design was used. Persons affected by leprosy, their close contacts, community members and health workers were included. Through interview-administered questionnaires we assessed knowledge, attitudes, practices and fears with the KAP measure, EMIC-CSS and SDS. In addition, semi-structured interviews and focus group discussions were conducted. The quantitative data were analysed using stepwise multivariate regression. Determinants of knowledge and stigma that were examined included age, gender, participant type, education, occupation, knowing someone affected by leprosy and district. The qualitative data were analysed using open, inductive coding and content analysis.

We administered questionnaires to 2344 participants (46% from India, 54% from Indonesia) as an interview. In addition, 110 participants were interviewed in-depth and 60 participants were included in focus group discussions. Knowledge levels were low in both countries: 88% of the participants in India and 90% of the participants in Indonesia had inadequate knowledge of leprosy. In both countries, cause, mode of transmission, early symptoms and contagiousness of leprosy was least known, and treatment and treatability of leprosy was best known. In both countries, health workers had the highest leprosy knowledge levels and community members the highest stigma levels (a mean score of up to 17.4 on the EMIC-CSS and 9.1 on the SDS). Data from the interviews indicated that people were afraid of being infected by leprosy. Local beliefs and misconceptions differed, for instance

information.org/resource/leprosy-perceptions-and-knowledge-endemic-districts-india-and-indonesia-differences-and.

**Funding:** This study is part of a larger research project, the Post-Exposure Prophylaxis (PEP++) project. The PEP++ project is funded by the Dream Fund of the Dutch Postcode Lottery. The funders had no role in study design, data collection and analysis, decision to publish, or preparation of the manuscript.

**Competing interests:** The authors have declared that no competing interests exist.

that leprosy is in the family for seven generations (Indonesia) or that leprosy is a result of karma (India). The determinants of leprosy knowledge and stigma explained 10–29% of the variability in level of knowledge and 3–10% of the variability in level of stigma.

## Conclusion

Our findings show the importance of investigating the perceptions regarding leprosy prior to educational interventions in communities: even though knowledge levels were similar, local beliefs and misconceptions differed per setting. The potential determinants we included in our study explained very little of the variability in level of knowledge and stigma and should be explored further. Detailed knowledge of local knowledge gaps, beliefs and fears can help tailor health education to local circumstances.

### Author summary

This study aimed to examine the differences and commonalities in and determinants of knowledge, attitudes, practices and fears regarding leprosy in endemic districts in India and Indonesia. Persons affected by leprosy, their close contacts, community members and health workers were included. We administered questionnaires (the KAP measure, EMIC-CSS and SDS) to 2344 participants. In addition, 110 participants were interviewed in-depth and 60 participants were included in focus group discussions. Knowledge levels were low. In both countries, cause, mode of transmission, early symptoms and contagiousness of leprosy was least known, and treatment and treatability of leprosy was best known. In both countries, health workers had the highest leprosy knowledge levels and community members the highest stigma levels. Data from the interviews indicated that people were afraid of being infected by leprosy. Local beliefs and misconceptions differed. The potential determinants we included in our study explained very little of the variability in level of knowledge and stigma and should be explored further. Our findings show the importance of investigating the perceptions regarding leprosy prior to educational interventions in communities: even though knowledge levels were similar, local beliefs and misconceptions differed per setting.

## Introduction

Many health conditions are associated with social stigma, including epilepsy, mental illness, disability and infectious diseases such as HIV/AIDS, tuberculosis, leprosy, lymphatic filariasis and Buruli ulcer. Stigma occurs when "elements of labelling, stereotyping, separation, status loss, and discrimination occur together in a power situation that allows them" [1]. Stigma is characterized by social exclusion or separation, rejection, blame and loss of status of an individual or group [1,2]. For many stigmatized individuals, the psychosocial consequences of their health condition are harder to bear than the physical consequences [3,4]. Stigma is associated with poor psychosocial health outcomes such as reduced quality of life, low self-esteem, depression and social exclusion [3,5]. This is also the case for persons affected by leprosy who experience stigma [6].

Leprosy is an infectious disease that primarily affects and damages the peripheral nerves and skin, which can result in disabilities [7,8]. Leprosy has had a very negative image for

hundreds of years and is known for being a very stigmatized condition [9]. Leprosy and its stigma may affect different areas of a person's life such as mobility, social relationships, marriage, employment and social participation [4]. Stigma and discrimination can lead to stress, anxiety, depression, suicide, isolation and problems in interpersonal relationships of persons affected [4]. Stigma in leprosy can also worsen already existing social inequalities due to age, gender and social class or status [10].

Stigma, as well as a lack of knowledge about leprosy, are obstacles to case finding and adherence to treatment [11,12] and therefore reduce the effectiveness of leprosy care and control activities [4,13]. In an attempt to hide their disease and prevent discrimination, stigmatized individuals often delay seeking treatment until they develop permanent, visible disabilities [8]. When people delay seeking treatment, transmission of the disease is prolonged, which hinders the treatment and prevention of the disease. To improve strategies for early case detection it is essential to improve the knowledge of leprosy and reduce stigma.

There are many factors that contribute to the stigma of leprosy, including fear of transmission and contagion, the visible manifestations such as deformity and disability in persons affected and religious and cultural beliefs regarding causes and treatment of leprosy [12,13]. Knowledge about leprosy plays a crucial role in stigma [14–18]. Local (mis)beliefs, such as the beliefs that all leprosy patients end up with disabilities, that leprosy is not curable or results in death or that imply that the person affected has done wrong and brought the disease upon himself all contribute to stigma [13,14,18–23]. Research showed that personal characteristics such as gender [15,16,22,24], occupation [16,22,24–26], years of education [15–17,22,25], age [15,25,27,28] and living area [15,24,27,29,30] are associated with community stigma against persons affected by leprosy.

Although knowledge, attitudes and practices regarding leprosy have been studied extensively and different determinants of knowledge and community stigma have been identified, we lack consensus about determinants of stigma and insight in how we can change negative perceptions and stigmatising local beliefs about leprosy. Insight in the dynamics, differences and commonalities in knowledge, attitudes and practices between leprosy endemic countries can help us to better target interventions to improve the knowledge and perception of leprosy, and thus reduce stigma. This study aims to examine the differences and commonalities in and determinants of knowledge, attitudes, practices and fears regarding leprosy in leprosy endemic districts in India and Indonesia.

## Methods

### Ethics statement

Ethical approval for this study was obtained as part of a larger research project: the Post-Exposure Prophylaxis project (PEP++ project). Ethical approval was obtained from the Institutional Ethics Committee in India and from Airlangga University in Indonesia. All participants were fully informed about the nature, objectives and procedures of the study, their rights and of confidentiality of the data prior to data collection. Written consent for participation in the study was obtained from each participant. For minors (participants below 18 years of age), verbal informed consent was obtained from the minor and written informed consent was obtained from one of the minor's legal representatives, e.g. a parent or guardian.

### Study design

The study used a cross-sectional research design with a mixed methods approach. Interviewer-administered questionnaires included demographic characteristics and knowledge and

attitudes of people towards (persons affected by) leprosy. In-depth information was obtained using semi-structured interviews and focus group discussions.

## Study site

The study was conducted in Fatehpur and Chandauli district, Uttar Pradesh, India and in Pamekasan and Pasuruan regencies (including Pasuruan city) in East Java, Indonesia. India and Indonesia are among the three most endemic countries for leprosy. India and Indonesia account for 92% of the South-East Asian region's case load and for almost 66% of the global disease burden of leprosy [31]. In March 2017, the prevalence of leprosy was 0.89 per 10.000 population in Fatehpur and 0.66 per 10,000 population in Chandauli district [32]. The prevalence of leprosy was 2.27 per 10,000 population in Pamekasan regency, 1.07 per 10,000 population in Pasuruan regency and 0.55 per 10,000 population in Pasuruan city in December 2018 respectively (data obtained from the local Provincial Health Offices).

## Study population and sample

Four groups of people were included in the study: (1) persons affected by leprosy; (2) close contacts of persons affected; (3) community members; and (4) health care workers. We aimed to include a random sample of at least 100 persons of each target group per country for the interview-administered questionnaires. In addition, in each country we aimed to have semi-structured interviews with six persons from each participant group and one focus group discussion per participant group. These participants were to be a subset of those in the quantitative sample. More information about the sample size calculation can be found elsewhere [18].

## Eligibility criteria

Participants needed to be inhabitants of one of the districts included in the study. Only persons affected diagnosed with leprosy within the last five years were included. Closest contacts had to have had intensive contact with the person affected for at least 20 hours per week for at least three months in the 12 months before the person affected was diagnosed. Close contacts included household contacts, family members, neighbours and other social contacts. Only those community members living in the same village or neighbourhood as the person affected by leprosy were asked to participate. Health care workers included professionals and volunteers, and persons with and without specific responsibilities for leprosy treatment services. Persons below the age of 16 and persons unwilling or unable to give informed consent were excluded. Close contacts, community members and health care workers were also excluded if they were or had ever been diagnosed with leprosy. Participants who were listed as close contact could not participate as community member also.

## Sampling methods

Participants for the interview-administered questionnaires were selected as follows:

1. The persons affected were selected by stratified systematic sampling with a random start from a list of leprosy patients registered at the primary health care centre.

2. In India, close contacts were selected by convenience sampling. We realized in hindsight that convenience sampling was not the best approach and therefore applied random sampling of close contacts when initiating the data collection in Indonesia. In Indonesia, all persons affected included in the study were each asked to name their 20 closest contacts whose names were written down on pieces of paper. These pieces were put in a cup, and one was

randomly drawn. If this person was not available or did not give consent, another name was randomly drawn. In both countries one close contact per person affected was included.

3. The community members were selected by convenience sampling from among those living in the same village or neighbourhood as the person affected by leprosy. We tried to select the community members as randomly as possible by selecting one or two community members per person affected from within a radius of 500 meters of the house of the person affected.

4. Health care workers were selected based on convenience sampling from among those present and available at the primary health care centres in the district. Half of the health care workers included in this study had received training about leprosy and had specific responsibilities for leprosy treatment services.

The participants for the qualitative interviews were a subset of those in the quantitative sample. Participants in the interviews were selected using purposive sampling to ensure adequate representation of age, sex and villages.

## Data collection

Three measures were used in this study: a knowledge, attitudes and practices (KAP) measure, the Explanatory Model Interview Catalogue Community Stigma Scale (EMIC-CSS) and the Social Distance Scale (SDS). Demographic information was also obtained from all participants. The EMIC-CSS and SDS were not administered to persons affected, since they assess community stigma.

The up-to 17-item (depending on the participant type) KAP measure was used to assess the knowledge, attitudes and practices of persons affected, close contacts, community members and health care workers regarding leprosy. Based on seven items on the KAP measure considering knowledge of leprosy a total knowledge score is calculated. A total score of eight could be obtained on the KAP adding up the correct answers even if incorrect answers were present. We defined 'poor knowledge' as a score of two or less out of eight on the KAP measure ($\leq$25% correct), 'moderate knowledge' as a score between two and six (25–75% correct) and 'adequate knowledge' as a score of six or more ($\geq$75% correct). The KAP measure has been used in several leprosy studies in Nepal, India, Indonesia and Brazil between 2012 and 2017 [18,33–35].

The 15-item EMIC-CSS was used to measure attitudes and behaviour towards persons affected by leprosy. The total score ranges from zero (no negative attitudes) to 30 (most negative attitudes). The EMIC-CSS has been validated among community members of persons affected by leprosy in India and Indonesia [36,37].

The 7-item SDS was used to assess the social distance the interviewee wants to keep towards persons affected by leprosy as a proxy for their attitudes. The SDS total score ranges from zero (no negative attitudes) to 21 (most negative attitudes). The SDS has been validated among community members of persons affected by leprosy in Indonesia [37]. The SDS has been translated to Hindi using forward and backward translation and partially validated (item interpretability, floor and ceiling effects and internal consistency) showing adequate validity, among community members of persons affected by leprosy in Uttar Pradesh, India [33].

Cross-sectional data on attitudes and perceptions of the participants towards (persons affected by) leprosy were also obtained using semi-structured in-depth interviews and focus group discussions. All participants were interviewed by a trained local interviewer at or near their home. The interview guide was pilot tested in each district before use, resulting in minor revisions to the guide. Participants of the pilot interviews were not included in the final sample. The in-depth interviews and focus group discussions were audio recorded, transcribed

verbatim and translated into English. Data were collected in different intervals between March 2017 and December 2018.

## Data analysis

Quantitative data analyses were performed in SPSS version 24. Simple descriptive methods were used to generate a demographic profile of the study sample. Stepwise multivariate regression with backward elimination was done to investigate the contribution of potential determinants (age, gender, participant type, education, occupation, knowing someone affected by leprosy and district) to the outcomes of interest (knowledge, stigma and social distance). Dependent variables that originally consisted of multiple categories were recoded into binary dummy variables. Independent variables were considered for entry into the multivariable logistic regression model if they had a $p$-value of $\leq 0.2$ obtained in univariate analysis. Variables with $p$-values of $\geq 0.05$ were eliminated one-by-one from the multivariate model until all variables that remained in the model were statistically significant ($p<0.05$). Bootstrapped stepwise multivariate regression with backward elimination, to correct for non-normality, was done for dependent variables that were distributed non-normally.

For each dependent variable four models were made: one using the whole database (four districts, all participant types), one for health workers only, one for persons affected only and one for the general population (close contacts and community members). Separate models for health workers were made because we considered them a more heterogenous group and a different group than the general population in terms of having completed (higher) education, knowledge (training) about leprosy and occupation. Separate models for persons affected were made because of their personal experience with leprosy and because they likely received a briefing or information about their condition. We hypothesized that participant type correlated with dependent and independent variables in the model. To control for confounding of participant type, and to ensure that the effects of participant type would be removed from the final results, we decided to always keep all participant types in the model, when analysing the whole dataset.

The recordings of interviews were transcribed, translated to English and analysed using open, inductive coding and content analysis. Similar phrases with recurring themes were clustered together in tables, to identify connections. Qualitative data analyses were performed in Nvivo version 12, Microsoft Word and Microsoft Excel. All records were anonymised before analysis.

## Results

### Demographic information

A total of 2344 participants were included. A little over half (n = 1277, 54%) of the participants were from Indonesia. The average age of the participants was 40.5 years. Approximately half of the participants were female (41% in India (n = 433), 51% in Indonesia (n = 654)). Four groups of people were included in the study: 19% of the participants were affected by leprosy (n = 438), 19% were close contacts of persons affected (n = 449), 54% were community members (n = 1256) and 9% health care workers (n = 201). An overview of all participant characteristics can be found in Table 1.

In addition, a total of 110 participants (52 in India, 58 in Indonesia) were interviewed in-depth (average age 39 years, 45% female) and in India 60 participants were included in seven focus group discussions (average age 40 years, gender not recorded). These 170 participants were a subset of those who had completed the questionnaires. An overview of all participant in the interviews can be found as supporting information file (S1 Table).

**Table 1. Socio-demographic characteristics and leprosy-related stigma and knowledge levels.**

|  | India ($n = 1067$)[a] | Indonesia ($n = 1277$)[b] | Total ($n = 2344$) |
|---|---|---|---|
| Average age (range; SD) | 40.4 (16–90; 15.6) | 40.5 (16–95; 13.0) | 40.5 (16–95; 14.2) |
| Female, $n$ (%) | 433 (40.6) | 654 (51.2) | 1087 (46.4) |
| Participant type, $n$ (%) |  |  |  |
| Person affected by leprosy | 200 (18.7) | 238 (18.5) | 438 (18.7) |
| Close contact of person affected | 211 (19.8) | 238 (18.5) | 449 (19.2) |
| Community member | 556 (52.1) | 700 (55.0) | 1256 (53.6) |
| Health care worker | 100 (9.4) | 101 (8.0) | 201 (8.6) |
| No education, $n$ (%) | 279 (26.1) | 264 (20.7) | 543 (23.2) |
| Religion, $n$ (%) |  |  |  |
| Islam | 89 (8.3) | 567 (45.1) | 1365 (28.4) |
| Hinduism | 970 (90.9) | 0 (0.0) | 970 (41.4) |
| Unknown | 8 (0.7) | 700 (54.8) | 708 (30.2) |
| Occupation, $n$ (%) |  |  |  |
| Paid work | 349 (32.7) | 250 (19.6) | 599 (25.6) |
| Self-employed | 347 (32.5) | 615 (48.2) | 962 (41.0) |
| Retired or unemployed | 82 (7.7) | 192 (15.0) | 274 (11.7) |
| Other (such as non-paid work, student) | 289 (27.1) | 220 (17.2) | 509 (21.7) |
| Questionnaire scores, mean |  |  |  |
| KAP measure (range 0–8)[c] | 3.9 | 3.2 | 3.5 |
| EMIC-CSS (range 0–30)[d] | 15.9 | 15.5 | 15.7 |
| SDS (range 0–21)[d] | 6.6 | 8.6 | 7.7 |

a 446 participants from Fatehpur and 621 participants from Chandauli district.

b 639 participants from Pamekasan and 638 participants from Pasuruan regency.

c In the presence of incorrect answers. Participants could give multiple answers to some of the KAP questions.

d The EMIC-CSS and SDS were administered to close contacts, community members and health workers.

## Differences and commonalities in knowledge, attitudes and practices regarding leprosy

**Differences and commonalities in leprosy knowledge and beliefs.** The questions related to knowledge that were answered correctly most and least frequently in both countries were the same. An overview of the number of correct responses given per participant group per country, per knowledge question of the KAP measure can be found as supporting information files (S2 Table and S1 Text). Mode of transmission, cause, early symptoms and whether leprosy is contagious after treatment or not was least known among all participant groups. In both countries treatment and treatability of leprosy was best known (>74% correct). In addition, in both countries health workers had significantly better knowledge than the other participants (independent samples t-test, p<0.001). When comparing overall scores between participant groups, merging the data from India and Indonesia, we found that persons affected had significantly higher knowledge scores than close contacts and community members, and that health workers had significantly higher knowledge scores than all participant groups (independent samples t-test, p<0.001).

Persons affected, contacts and community members from India had significantly higher mean knowledge scores (independent samples t-test, p<0.05) than the same participant groups in Indonesia. This considered e.g. questions about treatment and treatability (91–93% correct in India versus 75–78% in Indonesia), prevention of disabilities (69% correct in India

versus 50% Indonesia) and contagiousness after treatment (43% correct in India versus 23% in Indonesia). Participants from Indonesia had slightly better knowledge about mode of transmission (12% correct in Indonesia versus 6% in India) and cause of leprosy (17% correct in Indonesia versus 13% in India). While health workers from Indonesia had significantly better (p = 0.001) knowledge than health workers from India on almost all aspects, their knowledge about whether leprosy is contagious or not after treatment was low (9% correct in Indonesia versus 59% in India).

We found that 12% of the participants in India and 10% of the participants in Indonesia had adequate knowledge of leprosy. In addition, 76% of the participants in India and 58% of the participants in Indonesia had moderate knowledge and 14% of the participants in India and 32% of the participants in Indonesia had poor knowledge of leprosy.

Both on the KAP measure and during the in-depth interviews and focus group discussions, participants often used multiple explanations and held different beliefs per knowledge topic. This is illustrated by the following quote:

> "...[Being affected by leprosy] may be because of wearing wet clothes or some kind of allergy in my blood. It may also be that it had happened to some friend and I got infected while playing because it can spread through the touch..."–Person affected, male, India, in-depth interview

In addition, in both countries participants believed that certain types of food or drinks, for example seafood or unhealthy food, could cause leprosy. While some participants in both countries believed that an unclean environment could cause leprosy, this belief was more prominent in India. At the same time, some participants in both countries believed leprosy was hereditary, but this was a much more prominent belief in Indonesia. Some participants believed in supernatural causes of leprosy, especially in Chandauli in India (karma and evil spirits) and Pasuruan in Indonesia (black magic, God's will).

There were several local beliefs around the cause and mode of transmission of leprosy in Indonesia. Participants believed that leprosy is caused by 'impure blood' caused by 'karuwat sin', which creates 'bad flesh'. Bad flesh is also one of the names used for leprosy in Indonesia. Karuwat is when a man and a woman have sexual intercourse while the woman has her period and conceive a child. It is believed that the child will be affected by leprosy. One participant explained:

> "...To my knowledge, women in their menstrual cycle are not permitted to have sexual relations in any religion. So the majority of the community here believe that to be the cause. Bacteria is the cause, and it is believed that [babies] born carrying contaminated bacteria end up having leprosy..."

> - Close contact, male, Indonesia, in-depth interview

In addition, some participants in Indonesia believed that leprosy is in the family for seven generations. Another less common belief in Indonesia was that one can get leprosy from stepping on the grave of a person affected by leprosy. One participant explained:

> "...When [a friend and I] were in junior high school [we] were playing on the graveyard, cemetery, and [we] stepped on hot soil. Here hot means not hot literally (...) it is like contagious (...) the graveyard is from people with leprosy, so they get infected [by stepping on that land]

*(. . .) The one that stepped on it is the one that untreatable. But the other friend got it from hereditary. . .*"–Community member, male, Indonesia, in-depth interview

In addition, some participants in Indonesia believed there were two types of leprosy: 'lepra', the skin condition and '*kusta*', a more severe and feared form with visible impairments. Many participants in India thought leprosy transmits by touch and linked this to 'untouchability' and sometimes to being religiously unclean. One participant said:

"*. . .Some people maintain distance, they don't eat or drink together because of untouchability. . .*"

- Community member, female, India, in-depth interview

**Differences and commonalities in leprosy-related stigma.**   The mean EMIC-CSS (stigma) score was 15.9 in India and 15.5 in Indonesia. In both countries, community members had the highest mean EMIC-CSS score (17.4 India, 17.0 Indonesia), followed by health workers (15.0 India, 13.7 Indonesia) and close contacts (12.4 India, 11.8 Indonesia). Differences between the countries in mean EMIC-CSS scores per participant group were not significant (independent samples t-test, p>0.05, Table 2). However, when comparing overall scores between participant groups, merging the data from India and Indonesia, we found that close contacts had significantly lower and community members significantly higher mean EMIC-CSS scores (independent samples t-test, p<0.001).

The EMIC-CSS questions reflecting negative attitudes that were frequently endorsed (indicating most stigma) were similar between the two countries. These questions relate to marriage, avoiding persons affected, shame and disclosure. An overview of all responses to the EMIC-CSS can be found as supporting information file (S1 Fig).

The overall mean SDS score was 6.6 in India and 8.6 in Indonesia (p<0.05), indicating more stigma in Indonesia. In both countries community members had the highest mean SDS scores (7.2 India, 9.1 Indonesia), followed by close contacts (6.7 India, 8.6 Indonesia) and health workers (3.4 India, 5.3 Indonesia), see Table 2. Health workers had significantly lower SDS scores than close contacts and community members (independent samples t-test, p<0.001). In both countries, the questions indicating most stigma were the same and relate to having a person affected by leprosy as caretaker of one's children and to having one's children marry a person affected by leprosy (>45% negative responses in both countries). An overview of all responses to the SDS can be found as supporting information file (S1 Fig).

In the in-depth interviews and focus group discussions, many participants indicated that community members have a negative attitude towards persons affected by leprosy and avoid or exclude persons affected by leprosy. The main explanation in both countries was fear getting infected by the disease. One participant explained:

"*. . .I feel sorry for them. One, because they are alienated from their community. Two, because rarely ever would anyone talk to them or involve them or allow them to raise their own children. But what to do? As a community, if someone has leprosy, we fear for our own health, fear of infection. It is horrifying. . .*"—Close contact, female, Indonesia, in-depth interview

Especially in India, some participants had a very negative perception of persons affected. One participant said:

"*. . .Most leprosy patients are dirty and poor. So, if they don't treat their leprosy it will go on, go on, go on. It gets worse. If it gets worse, working is not possible. No job for them. No one*

**Table 2. Differences in total scores on the KAP (range 0–8), EMIC-CSS (range 0–30) and SDS (range 0–21) per participant group, per country.**

| Instrument and participant type | | Mean India (*n* = 1067) | Mean Indonesia (*n* = 1277) | p-value[b] |
|---|---|---|---|---|
| KAP[a] | Persons affected by leprosy | 3.9 | 3.4 | .000 |
| | Close contacts | 3.5 | 3.0 | .001 |
| | Community members | 3.8 | 2.8 | .000 |
| | Close contacts and community members[c] | 3.7 | 2.9 | .000 |
| | Health workers | 5.6 | 6.2 | .001 |
| | All groups | 3.9 | 3.2 | .000 |
| EMIC-CSS | Close contacts | 12.4 | 11.8 | .356 |
| | Community members | 17.4 | 17.0 | .268 |
| | Close contacts and community members | 16.0 | 15.6 | .289 |
| | Health workers | 15.0 | 13.7 | .217 |
| | All groups | 15.9 | 15.5 | .188 |
| SDS | Close contacts | 6.7 | 8.6 | .000 |
| | Community members | 7.2 | 9.1 | .000 |
| | Close contacts and community members | 7.0 | 9.0 | .000 |
| | Health workers | 3.4 | 5.3 | .002 |
| | All groups | 6.6 | 8.6 | .000 |

a In the presence of incorrect answers. Participants could give multiple answers to some of the KAP questions.

b p-value of the difference in total questionnaire score between India and Indonesia in the corresponding column. Tested using an independent samples t-test (significance, 2-tailed, equal variances were assumed if Levene's test for equality of variances had a p-value of >0.05).

c This is a merged group that combines the data of close contacts and community members.

*would want to work with leprosy patients, no one would like to have leprosy patients employed, no one would buy from leprosy patients. So, if leprosy is visible, they can't do work."*–Community member, female, India, in-depth interview

Another participant said:

*". . .People do not eat with leprosy patient nor touch them. And his living place is also separated. They refuse to visit some places and says don't touch me, or else I will also have disease. Because of this he becomes sad and suffers from inferiority complex. Sometimes he also tried to commit suicide. . ."*—Community member, gender unknown, India, focus group discussion

On the other hand, some participants said that there was no discrimination in the community. In addition, most participants in the in-depth interviews said that in general health workers have a good attitude towards persons affected by leprosy.

In Indonesia some participants stressed that when leprosy is not visible, the community will treat these individuals normally. It became clear in the interviews that in Indonesia the terminology around leprosy is sensitive. Some health workers indicated it's better not to tell someone they have leprosy and had therefore adopted different names:

*". . .In the Madurese context we should not mention the word leprosy; instead we refer to it as skin condition that is treatable. This is so that the patients do not evade treatment. If we do [say that it is leprosy] then patients definitely will not come back. . ."*–Health worker, male, Indonesia, in-depth interview

Some participants indicated that they used different names for '*kusta'* (leprosy), such as '*daging jubek'* or '*budukan'* (bad flesh) or '*gatal gatal'* (itchy). These terms are only used to indicate leprosy. In India participants indicated that they were reluctant to touch persons affected by leprosy.

### Determinants of leprosy knowledge and leprosy-related stigma

**Determinants of knowledge of leprosy.**    Multivariate analysis showed that being illiterate, not having had any (formal) education, only completing primary education, not knowing anyone affected by leprosy and living in Pamekasan or Pasuruan district (Indonesia) were all associated with lower levels of knowledge of leprosy (Table 3). Multivariate regression models per participant group could explain 11–22% of the variability of knowledge of leprosy, see Table 3. The determinants of knowledge of leprosy per participant group related to age, gender, knowing someone affected by leprosy, education and area of residence.

### Determinants of leprosy-related stigma

Multivariate analysis showed that living in Chandauli, Fatehpur or Pamekasan district was associated with higher levels of stigma towards persons affected by leprosy (Table 3). This model explained 10% of the variability of stigma. Models per participant group can be found in Table 3. The models per participant group showed that age was the only determinant of leprosy community stigma for health workers, and Chandauli, Fatehpur or Pamekasan district the only determinants of stigma in contacts and community members.

Female gender, not having completed higher education, low knowledge about leprosy, and living in Chandauli, Fatehpur or Pasuruan district were associated with higher levels of social distance towards persons affected by leprosy. This model explained 10% of the variability of social distance. Multivariate regression models per participant group can be found in Table 3. These models explained 3–7% of the variability of social distance towards persons affected by leprosy.

## Discussion

We found both in India and Indonesia that knowledge about leprosy was poor, while community stigma towards (persons affected by) leprosy was high. There were differences in levels of knowledge and stigma between participant groups: knowledge was better among health workers and stigma was higher among community members. The levels of knowledge were similar in both countries, but the explanations given, the 'local beliefs' and 'misconceptions', differed for some topics. Our findings identified three main drivers of stigma: (1) poor knowledge and misconceptions about leprosy, (2) local beliefs, and (3) fear of contagion. We will now discuss these findings in more detail by drivers of stigma and by participant group.

### Poor knowledge and misconceptions

Lower levels of knowledge of leprosy were associated with higher levels of social distance, a proxy for fear and stigma in the community. Lacking knowledge about leprosy is more often found to be associated with negative attitudes towards persons affected by leprosy [14–18]. Misconceptions such as that leprosy transmits by touch, a prominent belief among participants from India in the present study, increase stigma. These misconceptions are often linked to fear of the disease and fear of transmission [13,20,38,39]. To reduce stigma these misconceptions need to be addressed and challenged and knowledge needs to be increased. This is also crucial to improve strategies for early case detection, since lack of knowledge of leprosy is a major contributing factor to late diagnosis [7].

**Table 3. Correlations between level of knowledge (KAP score, in the presence of incorrect answers) about leprosy, community stigma (EMIC-CSS), social distance (SDS) and the other variables in the dataset per participant group.** Full models can be found as supporting information file (S1 Text).

| | Determinants of <u>lower</u> levels of knowledge (KAP measure) | R-squared | Determinants of <u>higher</u> levels of stigma (EMIC-CSS) | R-squared | Determinants of <u>higher</u> levels of social distance (SDS) | R-squared |
|---|---|---|---|---|---|---|
| Whole dataset | • Illiterate, no (formal) education or only completed primary education<br>• Does not know someone affected by leprosy<br>• From Pamekasan or Pasuruan district | 0.286 | • From Chandauli, Fatehpur or Pamekasan district | 0.096 | • Female gender<br>• Not having completed higher education<br>• Lower knowledge about leprosy<br>• From Fatehpur, Pamekasan or Pasuruan district | 0.103 |
| Persons affected | • Older age<br>• Female gender<br>• Not completed any (formal) education<br>• From Pamekasan district | 0.120 | - | - | - | - |
| Contacts and community | • Illiterate, no (formal) education or only completed primary education<br>• Does not know someone affected by leprosy<br>• From Pasuruan or Pamekasan district | 0.107 | • From Chandauli, Fatehpur or Pamekasan district | 0.103 | • Female gender<br>• Not having completed higher education<br>• Lower knowledge about leprosy<br>• From Fatehpur, Pamekasan or Pasuruan district | 0.069 |
| Health workers | • Female gender<br>• Does not know someone affected by leprosy<br>• From Pamekasan, Fatehpur or Chandauli district | 0.237 | • Younger age | 0.032 | • Older age | 0.033 |

## Local beliefs

Interestingly, even though the questions related to knowledge that were answered correctly most and least frequently were the same in both countries in the present study, the local beliefs, especially considering the cause and mode of transmission, varied by area of residence. This confirms findings from other studies that showed that (socio)cultural beliefs about leprosy can increase stigma [13,14,18–23]. We found several local beliefs that can be addressed, such as the belief that leprosy is in the family for seven generations, that a cause of leprosy is that a woman conceives while having sexual intercourse during her period (Indonesia), that leprosy has a supernatural cause and that persons affected by leprosy are untouchable (India).

Some studies have suggested that these beliefs are influenced by religious beliefs and religious teachings about leprosy [13,19,20,40]. We hypothesize that the local beliefs in the present study have to some extent also been influenced by religion and religious practices. For example in Indonesia, where almost all participants were Muslim, local beliefs regarding the cause of leprosy revolved around stepping on a grave (people are buried in Islam and Christianity but cremated in Hinduism) and sexual intercourse with menstruating women (explicitly mentioned as prohibited in the Quran). In India, where almost all participants were Hindu, local beliefs revolved around karma (a fundamental concept of Hinduism), untouchables/untouchability (the former name for a member of low-caste Hindu groups) and being religiously unclean (untouchables are considered religiously unclean).

## Fear of contagion

A third important driver of stigma found in the present study was that people were afraid of getting infected with the disease. This is something found in other studies also [13,20,38,39] and something that should receive specific attention when designing leprosy campaigns.

## Persons affected by leprosy

We found that persons affected had significantly better knowledge of leprosy than close contacts and community members. This is similar to findings from a study in India, that found that persons affected by leprosy had higher knowledge scores than their family members [17]. Persons affected likely have better knowledge because of their personal experience with the disease and because they have often received information about their condition from health workers when they were diagnosed. However, knowledge about leprosy was low in the present study. Similar to our findings, several other studies in India reported low or inadequate levels of knowledge of leprosy among persons affected [41–43]. Low levels of leprosy knowledge may contribute to non-compliance to treatment and need to be addressed [44].

## Close contacts of persons affected and community members

The present study found that community members had the highest stigma levels of all participant groups. This may be explained by their poor knowledge about leprosy, something that has been associated with higher levels of stigma towards persons affected by leprosy in other studies also [14–17]. The image that community members have of persons affected by leprosy is likely not based on knowledge from personal contact, but on incorrect information and negative beliefs.

The present study reported mean stigma scores (EMIC-CSS) ranging from 11.8 (contacts) to 17.4 (community members), which is above the cut-off score for perceived stigmatisation of 8, as proposed by Sermrittirong and colleagues [45]. This confirms findings in Indonesia, Brazil, Thailand, Nepal, Nigeria and New Zealand (mean or median EMIC-CSS scores ranging from 12 to 18) [14,16,22,25,28,45–48].

Desired social distance towards persons affected by leprosy, how close one is willing to be towards an affected person in a given situation, is an indicator of the fear and attitude of the respondent themselves and a proxy for fear and stigma in the community. In Indonesia, social distance was assessed among community members in two studies using the SDS [37,46]. The mean SDS scores of 9.3 and 10.5 found in the SARI Project [37,46] are very similar to the score of 9.1 we found among community members in Indonesia, indicating a similar desire for social distance in the present study. These results indicate the desire of community members to keep a distance from persons affected by leprosy. Interestingly, while close contacts had a much lower mean perceived community stigma (EMIC-CSS) score, their mean social distance (SDS) score was about the same as that of community members. We expect that the difference between stigma and social distance scores of close contacts can be explained by the way the questions are asked. In the EMIC-CSS respondents are asked how 'others' feel or behave, while in the SDS respondents are asked how they themselves would feel relating to the person portrayed in a vignette. Thus, the SDS assesses personal attitudes and fears and the EMIC-CSS perceived attitudes and behaviour of others.

Several determinants of stigma have been identified in other studies, including knowledge of leprosy [14–18], (cultural) beliefs [13,14,18–23], female gender [15,16,22,24], occupation [16,22,24–26], fewer years of education [15–17,22,25], older age [15,25,27,28], knowing a person affected [28], religious beliefs [13,19,20,40] and living area [15,24,27,29,30]. We included almost all of these determinants, except for living area and religion, and found that together they explained very little of the variability in level of stigma (7% on the EMIC-CSS and 10% on the SDS).

We expect that 'local beliefs' and local explanations play an important role in knowledge and stigma and that these explanations vary by area of residence. Furthermore, some studies have found additional determinants of stigma, such as having seen a leprosy patient [21],

regulations regarding leprosy [20], exposure to leprosy health promotion messages [13,19], marital status [16,22], economic status [24], ethnicity [14,16,28], distance of residence from the hospital [14] and migrant status [28]. We recommend including these variables in future studies, to get a better understanding of the underlying mechanisms behind stigma and knowledge. In addition, we think it would be helpful to explore the influence of religion better, not just taking beliefs, but also level of religious faith and dedication into account.

### Health workers

We found that in both countries health workers had the highest leprosy knowledge levels. This is likely due to their (para)medical training, while some had also received leprosy training. Having had more training and/or more years of service has been associated with better knowledge about leprosy among health workers in other studies also [49–53]. In addition, all health workers had completed higher education, which was associated with higher levels of knowledge about leprosy in all four participant groups in our study. Even though health workers had higher levels of knowledge than the other participant groups, many still lacked knowledge, for example about mode of transmission and contagiousness of leprosy. Standard training on leprosy for all health workers and regular refresher courses could likely improve this. Interestingly, the topics on which health workers lacked knowledge (e.g. mode of transmission and contagiousness after treatment) and had adequate knowledge (e.g. treatment) was reflected in the knowledge about leprosy among the other participant groups. This likely shows that adequate knowledge about a particular topic among health workers enables them to pass on correct information to persons affected, contacts and community members. In addition, we believe it reflects the messages that have been used in past government education campaigns. These findings underline the importance of ensuring that health workers have correct knowledge about leprosy.

The present study also found that health workers had high mean stigma scores of 13.7 (Indonesia) and 15.0 (India) on the EMIC-CSS. Determinants of stigma for health workers only included age, this only explained 3% of the variability in level of stigma. Determinants of stigma among health workers should be explored further. Because health workers are in a respected position in the community, their attitudes and behaviour can influence how others perceive leprosy. Health workers are therefore an important group to target with stigma reduction interventions.

### Interventions to improve the knowledge and perception of leprosy

Our findings indicate the need for effective interventions to positively influence the perception of leprosy and improve knowledge of leprosy. We believe our findings of local differences in knowledge gaps, misconceptions, beliefs and fears indicate that interventions should be culture-specific and contextualised [54,55]. This is expected to be much more effective to increase positive attitudes and acceptance of persons affected by leprosy than generic messages [40]. We believe our knowledge findings indicate that certain topics should be prioritized in health education in both countries: cause, mode of transmission, early symptoms and contagiousness of leprosy. These findings also show that some messages may be important as such, but do not have to be prioritized at the moment: knowledge about treatability of leprosy was good in both India and Indonesia. This is likely a reflection of the messages in past government education campaigns. While knowledge gaps can be addressed by information, attitudes, beliefs and fears require an additional approach. Changing knowledge and perceptions is best done as a combination of health education and behavioural change interventions [56,57].

Health education should target the general community, who had the highest stigma levels in both countries. This may be done by targeting key influencers and authority figures in the community, for example village leaders, who could influence others in the community and allow for the information to filter down.

## Strengths and limitations

One of the limitations of this study is its cross-sectional design, which prevented us from making more definitive causal inferences. Another limitation of this study is the difference in sampling methods of close contact selection in India and Indonesia. Close contacts in India were selected by convenience sampling, while close contacts in Indonesia were selected by random sampling. Despite differences in recruitment methods, the patterns of results showed parallels. Study strengths include the mixed-method approach that allowed for triangulation of the data.

## Conclusion

Our study revealed poor knowledge regarding leprosy in India and Indonesia, especially regarding cause, mode of transmission, early symptoms and contagiousness of leprosy. Knowledge about treatment and treatability was good. Stigma levels were high in both countries and were driven by poor knowledge and misconceptions about leprosy, local beliefs, and fear of contagion. These findings show the importance of investigating the perceptions regarding leprosy in the communities targeted for educational interventions. Local misconceptions and beliefs, especially around the cause and mode of transmission of leprosy, differed in the two countries. Contextualised health education and behaviour change interventions are required to improve knowledge, reduce misconceptions and positively influence the perception of leprosy. Interventions should address specific knowledge gaps, beliefs and fears.

The determinants of leprosy knowledge and stigma explained only a small proportion of the variability in level of knowledge and stigma. Future studies should attempt to find additional determinants to get a better understanding of the underlying mechanisms behind stigma and knowledge and find ways to improve interventions. We also recommend that future studies explore the role of religion, religiosity and area of residence in local beliefs.

## Supporting information

**S1 STROBE Checklist.**
(DOC)

**S1 Table. Number of participants included in the study, per country and per participant group.**
(DOCX)

**S2 Table. An overview of the number of correct responses given per participant group per country, per knowledge question of the KAP measure.**
(DOCX)

**S1 Fig. Figures EMIC and SDS.** The percentage of participants in India and Indonesia, indicating negative attitudes on the EMIC-CSS and SDS. The percentages are displayed as percentage of participants who answered "yes" (EMIC-CSS) or "definitely not willing" or "probably not willing" (SDS) on each question, out of all participants.
(DOCX)

**S1 Text. Regression models.** Regression models for the correlations between level of knowledge about leprosy (KAP measure), community stigma (EMIC-CSS), social distance (SDS)

and the other variables in the dataset.
(DOCX)

**S2 Text. An overview of the number of correct responses given per participant group per country, per knowledge question of the KAP measure.** The results are displayed by participant group, as percentage of participants who gave the correct answer.
(DOCX)

## Acknowledgments

We are grateful to the contributions of all of the participants who trusted us with their stories. We thank the research assistants who collected the data for the study. We would like to thank the principal investigators of the PEP++ project, Prof Jugal Kishore and Prof Cita Rosita Prakoeswa, and the NLR country directors, Dr Ashok Agarwal and Dr Asken Sinaga, for their support. We are grateful to the technical support of Dr Teky Budiawan of NLR Indonesia, and Prof Koesbardiati Toetik of Universitas Airlangga. We are especially grateful to Prof Jan Hendrik Richardus of Erasmus MC, University Medical Center, and Mr Duane Hinders, NLR, who supported and guided us in this study. We are grateful to Dr Daan Nieboer from Erasmus MC, University Medical Center, who provided statistical advice and support. Finally, we would like to thank Ms Aranka Ballering and Ms Kimriek Schutten-de Wild who helped collect data in Chandauli and Pasuruan district.

## Author Contributions

**Conceptualization:** Anna Tiny van't Noordende, Willem Herman van Brakel.

**Data curation:** Anna Tiny van't Noordende.

**Formal analysis:** Anna Tiny van't Noordende.

**Funding acquisition:** Willem Herman van Brakel.

**Investigation:** Anna Tiny van't Noordende, Suchitra Lisam, Panca Ruthindartri, Atif Sadiq, Vivek Singh, Miftahol Arifin, Willem Herman van Brakel, Ida J. Korfage.

**Methodology:** Anna Tiny van't Noordende, Willem Herman van Brakel, Ida J. Korfage.

**Project administration:** Suchitra Lisam, Panca Ruthindartri, Atif Sadiq, Vivek Singh, Miftahol Arifin.

**Resources:** Suchitra Lisam, Panca Ruthindartri, Atif Sadiq, Vivek Singh, Miftahol Arifin.

**Software:** Anna Tiny van't Noordende, Willem Herman van Brakel.

**Supervision:** Suchitra Lisam, Panca Ruthindartri, Atif Sadiq, Vivek Singh, Miftahol Arifin, Willem Herman van Brakel, Ida J. Korfage.

**Validation:** Anna Tiny van't Noordende.

**Visualization:** Anna Tiny van't Noordende.

**Writing – original draft:** Anna Tiny van't Noordende.

**Writing – review & editing:** Anna Tiny van't Noordende, Suchitra Lisam, Panca Ruthindartri, Atif Sadiq, Vivek Singh, Miftahol Arifin, Willem Herman van Brakel, Ida J. Korfage.

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
