## [Decision Letter · Decision Letter 0]

20 Oct 2020

Dear Ms. van 't Noordende,

Thank you very much for submitting your manuscript "Leprosy perceptions and knowledge in endemic districts in India and Indonesia: differences and commonalities" for consideration at PLOS Neglected Tropical Diseases. As with all papers reviewed by the journal, your manuscript was reviewed by members of the editorial board and by several independent reviewers. In light of the reviews (below this email), we would like to invite the resubmission of a significantly-revised version that takes into account the reviewers' comments. 

We cannot make any decision about publication until we have seen the revised manuscript and your response to the reviewers' comments. Your revised manuscript is also likely to be sent to reviewers for further evaluation.

Sincerely,

Susilene Maria Tonelli Nardi, Ph.D

Deputy Editor

Reviewer's Responses to Questions

**Key Review Criteria Required for Acceptance?**

**Methods**

-Are the objectives of the study clearly articulated with a clear testable hypothesis stated?

-Is the study design appropriate to address the stated objectives?

-Is the population clearly described and appropriate for the hypothesis being tested?

-Is the sample size sufficient to ensure adequate power to address the hypothesis being tested?

-Were correct statistical analysis used to support conclusions?

-Are there concerns about ethical or regulatory requirements being met?

Reviewer #1: In general, the methodology is well described. However, the authors should give more details about the focus group.

Also, authors should include information on how minors' consent was obtained.

Reviewer #2: Data analysis is adequate for the study proposal.

Reviewer #3: Gaps and problems statements/hypothesis / are absent. the gap should be indicated since there are many researches related to this study many where.

 The objectives of the study were clearly articulated. The population were clearly described though how of managing them seem vague. The sample size seem representative though there were discrepancy between two countries in sample selection . This should be justified. Statistical procedures were met and the significance level were clearly indicated. I trusted the authors and PLOS NEGLECTED TROPICAL DISEASES that I did not find attachment of Ethical clearance Certificate. I believe it should be there. Thank you.

**Results**

-Does the analysis presented match the analysis plan?

-Are the results clearly and completely presented?

-Are the figures (Tables, Images) of sufficient quality for clarity?

Reviewer #1: The authors mention that they carried out a focus group, however, the article does not specify the profile of the participants or bring the ideas and insights raised in the group interaction.

Reviewer #2: remove figure 1. It contains too much data making it impossible to interpret the information (line 297).

Reviewer #3: The analysis presented match the analysis plan that the KAP ( variables to be investigated were presented with the determinants). The reviewer found the results were clearly and completely presented from the perspectives of the objectives, and they were supported with figures and tables. Still there are areas which need validation regarding commonalities and differences in cases of neighborhood and distant communities' attitude. To be crosschecked... because it seemed generalization without clear consideration. I my self have a brother who is affected by leprosy. People in the distant area consider him as a unique and disregard from any participation. However the closer communities do not. It was due to that I conducted a communication research which mainly focuses on discriminatory perception, attitude and discourse. So, Completely I disagree that the neighbors have negative attitude towards leprosy affected people. Except this, every thing here is very fine. I am privileged to be assigned as a reviewer of this prestigious work.

**Conclusions**

-Are the conclusions supported by the data presented?

-Are the limitations of analysis clearly described?

-Do the authors discuss how these data can be helpful to advance our understanding of the topic under study?

-Is public health relevance addressed?

Reviewer #1: The conclusions are based on the results presented. The authors present the limitations and relevance of the study.

Knowing that health professionals are those who promote health education among affected individuals, contacts, and the community, it is interesting to note that the lack of knowledge about a particular topic among professionals is reflected in erroneous knowledge among affected individuals, contacts, and community members. In the same way that adequate knowledge on a given topic enables the professional to pass on correct information, as can be seen in the results shown in the S5 Figure. I suggest that the authors address these results in the discussion.

Reviewer #2: Study is relevant to create educational strategies focused on public health with a focus on controlling and fighting leprosy. Often the knowledge of individuals affected by leprosy is ignored, and many studies address clinical and epidemiological characteristics. This study in question brought more information beyond just numbers, it also brought people's speeches, their culture, their beliefs.

Reviewer #3: The public health relevance were addressed mentioning the number of leprosy case per10000 population in both countries. The issue is very convincing and it is very mandatory to work towards intervention by considering the determinant factors. I little bit mentioned in the main manuscript the presence of vagueness of data presentation to show how much they clearly indicate saying "Does your data suggest cause of differences in knowledge, attitude and perception or belief. If so, elaborating a little bit about it may make the information very complete" to increase clarity. In fact all the concluding points were emanating from the data presented.

**Editorial and Data Presentation Modifications?**

Reviewer #1: I suggest the authors eliminate unnecessary spaces between paragraphs during the text.

Line 348 - insert the numbering for the topic.

Reviewer #2: Confirm whether the number of individuals interviewed was 110 or 100. 

Why the selection of contacts from individuals in India was different from contacts in Indonesia.

Reviewer #3: I am really happy that this work needs very minor modification. Firstly, there should be theoretical foundation of the study(I have not noticed ). Secondly, the two countries may have documents to prevent negative views, bring about common knowledge or any other concerns which should support the study. I found the study well triangulated but still if things related to the problems are not abide by legal, or social rules people may not consider it forbidden to disregard leprosy affected people. So, would the authors consider this?

**Summary and General Comments**

Reviewer #1: It is a very interesting study that advances in the discussion about the determinants for lack of knowledge, stigma, and prejudice related to leprosy. Also, the authors emphasize the need for including new variables that could contribute to better explain these phenomena.

The study aims to examine the differences and commonalities in the determinants for knowledge, attitudes, and practices related to leprosy (lines 102-104). However, I believe that the study goes further by bringing results that also address stigma and prejudice, so I suggest that the authors review the objective of the study.

Reviewer #2: in the introduction of the text much is said about stigma and the other elements that will also be addressed in the study are not highlighted. I suggest that in addition to stigma, also mention more about social determinants and knowledge.

Reviewer #3: I would like from the strength of the study. It is very comprehensive that it includes the population from two countries. Secondly , the methodology is comprehensive that it is mixed. Data collection tools were clearly described and analysis were to the point. The weakness were not this much intensive. Theoretical foundations and gap to be field were not well explained. Differences were overlooked.

PLOS authors have the option to publish the peer review history of their article (what does this mean?). If published, this will include your full peer review and any attached files.

Reviewer #1: No

Reviewer #2: No

Reviewer #3: Yes: My Name is Dr Daniel Taye Feyisa. I am an Assistant Professor of Applied Linguistics and Communication. My PhD specialization is 'Health communication'. My PhD study is ' Analysis of Leprosy Discriminatory Discourses: Ethiopian leprosy affected people and their families. Thus, I have a very Solid background on the article I have reviewed here. Thank you for the opportunity.
---

## [Editor Report · Decision Letter 1]

2 Dec 2020

Dear Ms. van 't Noordende,

We are pleased to inform you that your manuscript 'Leprosy perceptions and knowledge in endemic districts in India and Indonesia: differences and commonalities' has been provisionally accepted for publication in PLOS Neglected Tropical Diseases.

Best regards,

Susilene Maria Tonelli Nardi, Ph.D

Deputy Editor

Susilene Nardi

Deputy Editor

---

## [Editor Report · Acceptance letter]

15 Jan 2021

Dear Ms. van 't Noordende,

We are delighted to inform you that your manuscript, "Leprosy perceptions and knowledge in endemic districts in India and Indonesia: differences and commonalities," has been formally accepted for publication in PLOS Neglected Tropical Diseases.

Best regards,

Shaden Kamhawi

co-Editor-in-Chief

Paul Brindley

co-Editor-in-Chief
